# Constructing Unrestricted Adversarial Examples with Generative Models

**Yang Song**
Stanford University
yangsong@cs.stanford.edu

**Rui Shu**
Stanford University
ruishu@cs.stanford.edu

**Nate Kushman**
Microsoft Research
nkushman@microsoft.com

**Stefano Ermon**
Stanford University
ermon@cs.stanford.edu

## Abstract

Adversarial examples are typically constructed by perturbing an existing data point within a small matrix norm, and current defense methods are focused on guarding against this type of attack. In this paper, we propose unrestricted adversarial examples, a new threat model where the attackers are not restricted to small norm-bounded perturbations. Different from perturbation-based attacks, we propose to synthesize unrestricted adversarial examples entirely from scratch using conditional generative models. Specifically, we first train an Auxiliary Classifier Generative Adversarial Network (AC-GAN) to model the class-conditional distribution over data samples. Then, conditioned on a desired class, we search over the AC-GAN latent space to find images that are likely under the generative model and are misclassified by a target classifier. We demonstrate through human evaluation that unrestricted adversarial examples generated this way are legitimate and belong to the desired class. Our empirical results on the MNIST, SVHN, and CelebA datasets show that unrestricted adversarial examples can bypass strong adversarial training and certified defense methods designed for traditional adversarial attacks.

## 1 Introduction

Machine learning algorithms are known to be susceptible to adversarial examples: imperceptible perturbations to samples from the dataset can mislead cutting edge classifiers [1, 2]. This has raised concerns for safety-critical AI applications because, for example, attackers could use them to mislead autonomous driving vehicles [3, 4, 5] or hijack voice controlled intelligent agents [6, 7, 8].

To mitigate the threat of adversarial examples, a large number of methods have been developed. These include augmenting training data with adversarial examples [2, 9, 10, 11], removing adversarial perturbations [12, 13, 14], and encouraging smoothness for the classifier [15]. Recently, [16, 17] proposed theoretically-certified defenses based on minimizing upper bounds of the training loss under worst-case perturbations. Although inspired by different perspectives, a shared design principle of current defense methods is to make classifiers more robust to small perturbations of their inputs.

In this paper, we introduce a more general attack mechanism where adversarial examples are constructed entirely from scratch instead of perturbing an existing data point by a small amount. In practice, an attacker might want to change an input significantly while not changing the semantics. Taking traffic signs as an example, an adversary performing perturbation-based attacks can draw graffiti [4] or place stickers [18] on an existing stop sign in order to exploit a classifier. However, the attacker might want to go beyond this and replace the original stop sign with a new one that was specifically manufactured to be adversarial. In the latter case, the new stop sign does not have

to be a close replica of the original one—the font could be different, the size could be smaller—as long as it is still identified as a stop sign by humans. We argue that *all inputs that fool the classifier without confusing humans can pose potential security threats*. In particular, we show that previous defense methods, including the certified ones [16, 17], are not effective against this more general attack. Ultimately, we hope that identifying and building defenses against such new vulnerabilities can shed light on the weaknesses of existing classifiers and enable progress towards more robust methods.

Generating this new kind of adversarial example, however, is challenging. It is clear that adding small noise is a valid mechanism for generating new images from a desired class—the label should not change if the perturbation is small enough. How can we generate completely new images from a given class? In this paper, we leverage recent advances in generative modeling [19, 20, 21]. Specifically, we train an Auxiliary Classifier Generative Adversarial Network (AC-GAN [20]) to model the set of legitimate images for each class. Conditioned on a desired class, we can search over the latent code of the generative model to find examples that are mis-classified by the model under attack, even when protected by the most robust defense methods available. The images that successfully fool the classifier without confusing humans (verified via Amazon Mechanical Turk [22]) are referred to as *Unrestricted Adversarial Examples*[1]. The efficacy of our attacking method is demonstrated on the MNIST [25], SVHN [26], and CelebA [27] datasets, where our attacks uniformly achieve over 84% success rates. In addition, our unrestricted adversarial examples show moderate transferability to other architectures, reducing by 35.2% the accuracy of a black-box certified classifier (i.e. a certified classifier with an architecture unknown to our method) [17].

## 2 Background

In this section, we review recent work on adversarial examples, defense methods, and conditional generative models. Although adversarial examples can be crafted for many domains, we focus on image classification tasks in the rest of this paper, and will use the words "examples" and "images" interchangeably.

**Adversarial examples**    Let $x \in \mathbb{R}^m$ denote an input image to a classifier $f : \mathbb{R}^m \to \{1, 2, \cdots, k\}$, and assume the attacker has full knowledge of $f$ (a.k.a., white-box setting). [1] discovered that it is possible to find a slightly different image $x' \in \mathbb{R}^m$ such that $\|x' - x\| \leq \epsilon$ but $f(x') \neq f(x)$, by solving a surrogate optimization problem with L-BFGS [28]. Different matrix norms $\|\cdot\|$ have been used, such as $l_\infty$ ([9]), $l_2$ ([29]) or $l_0$ ([30]). Similar optimization-based methods are also proposed in [31, 32]. In [2], the authors observe that $f(x)$ is approximately linear and propose the Fast Gradient Sign Method (FGSM), which applies a first-order approximation of $f(x)$ to speed up the generation of adversarial examples. This procedure can be repeated several times to give a stronger attack named Projected Gradient Descent (PGD [10]).

**Defense methods**    Existing defense methods typically try to make classifiers more robust to small perturbations of the image. There has been an "arms race" between increasingly sophisticated attack and defense methods. As indicated in [33], the strongest defenses to date are adversarial training [10] and certified defenses [16, 17]. In this paper, we focus our investigation of unrestricted adversarial attacks on these defenses.

**Generative adversarial networks (GANs)**    A GAN [19, 20, 34, 35, 36] is composed of a generator $g_\theta(z)$ and a discriminator $d_\phi(x)$. The generator maps a source of noise $z \sim P_z$ to a synthetic image $x = g_\theta(z)$. The discriminator receives an image $x$ and produces a value $d_\phi(x)$ to distinguish whether it is sampled from the true image distribution $P_x$ or generated by $g_\theta(z)$. The goal of GAN training is to learn a discriminator to reliably distinguish between fake and real images, and to use this discriminator to train a good generator by trying to fool the discriminator. To stabilize training, we use the Wasserstein GAN [34] formulation with gradient penalty [21], which solves the following optimization problem

$$\min_\theta \max_\phi \mathbb{E}_{z \sim P_z}[d_\phi(g_\theta(z))] - \mathbb{E}_{x \sim P_x}[d_\phi(x)] + \lambda \, \mathbb{E}_{\tilde{x} \sim P_{\tilde{x}}}[(\|\nabla_{\tilde{x}} d_\phi(\tilde{x})\|_2 - 1)^2],$$

where $P_{\tilde{x}}$ is the distribution obtained by sampling uniformly along straight lines between pairs of samples from $P_x$ and generated images from $g_\theta(z)$.

To generate adversarial examples with the intended semantic information, we also need to control the labels of the generated images. One popular method of incorporating label information into the generator is Auxiliary Classifier GAN (AC-GAN [20]), where the conditional generator $g_\theta(z, y)$ takes label $y$ as input and an auxiliary classifier $c_\psi(x)$ is introduced to predict the labels of both training and generated images. Let $c_\psi(y \mid x)$ be the confidence of predicting label $y$ for an input image $x$. The optimization objectives for the generator and the discriminator, respectively, are:

$$\min_\theta -\mathbb{E}_{z \sim P_z, y \sim P_y}[d_\phi(g_\theta(z, y)) - \log c_\psi(y \mid g_\theta(z, y)))]$$

$$\min_{\phi, \psi} \mathbb{E}_{z \sim P_z, y \sim P_y}[d_\phi(g_\theta(z, y))] - \mathbb{E}_{x \sim P_x}[d_\phi(x)] - \mathbb{E}_{x \sim P_x, y \sim P_{y|x}}[\log c_\psi(y \mid x)]$$
$$+ \lambda \, \mathbb{E}_{\tilde{x} \sim P_{\tilde{x}}}[(\|\nabla_{\tilde{x}} d_\phi(\tilde{x})\|_2 - 1)^2],$$

where $d_\phi(\cdot)$ is the discriminator, $P_y$ represents a uniform distribution over all labels $\{1, 2, \cdots, k\}$, $P_{y|x}$ denotes the ground-truth distribution of $y$ given $x$, and $P_z$ is chosen to be $\mathcal{N}(0, 1)$ in our experiments.

## 3 Methods

### 3.1 Unrestricted adversarial examples

We start this section by formally characterizing perturbation-based and unrestricted adversarial examples. Let $\mathcal{I}$ be the set of all digital images under consideration. Suppose $o : \mathcal{O} \subseteq \mathcal{I} \to \{1, 2, \cdots, K\}$ is an oracle that takes an image in its domain $\mathcal{O}$ and outputs one of $K$ labels. In addition, we consider a classifier $f : \mathcal{I} \to \{1, 2, \cdots, K\}$ that can give a prediction for any image in $\mathcal{I}$, and assume $f \neq o$. Equipped with those notations, we can provide definitions used in this paper:

**Definition 1** (Perturbation-Based Adversarial Examples). *Given a subset of (test) images $\mathcal{T} \subset \mathcal{O}$, small constant $\epsilon > 0$, and matrix norm $\|\cdot\|$, a perturbation-based adversarial example is defined to be any image in $\mathcal{A}_p \triangleq \{x \in \mathcal{O} \mid \exists x' \in \mathcal{T}, \|x - x'\| \leq \epsilon \wedge f(x') = o(x') = o(x) \neq f(x)\}$.*

In other words, traditional adversarial examples are based on perturbing a correctly classified image in $\mathcal{T}$ so that $f$ gives an incorrect prediction, according to the oracle $o$.

**Definition 2** (Unrestricted Adversarial Examples). *An unrestricted adversarial example is any image that is an element of $\mathcal{A}_u \triangleq \{x \in \mathcal{O} \mid o(x) \neq f(x)\}$.*

In most previous work on perturbation-based adversarial examples, the oracle $o$ is implicitly defined as a black box that gives ground-truth predictions (consistent with human judgments), $\mathcal{T}$ is chosen to be the test dataset, and $\|\cdot\|$ is usually one of $l_\infty$ ([9]), $l_2$ ([29]) or $l_0$ ([30]) matrix norms. Since $o$ corresponds to human evaluation, $\mathcal{O}$ should represent all images that look realistic to humans, including those with small perturbations. The past work assumed $o(x)$ was known by restricting $x$ to be close to another image $x'$ which came from a labeled dataset. Our work removes this restriction, by using a high quality generative model which can generate samples which, with high probability, humans will label with a given class. From the definition it is clear that $\mathcal{A}_p \subset \mathcal{A}_u$, which means our proposed unrestricted adversarial examples are a strict generalization of traditional perturbation-based adversarial examples, where we remove the small-norm constraints.

### 3.2 Practical unrestricted adversarial attacks

The key to practically producing unrestricted adversarial examples is to model the set of legitimate images $\mathcal{O}$. We do so by training a generative model $g(z, y)$ to map a random variable $z \in \mathbb{R}^m$ and a label $y \in \{1, 2, \cdots, K\}$ to a legitimate image $x = g(z, y) \in \mathcal{O}$ satisfying $o(x) = y$. If the generative model is ideal, we will have $\mathcal{O} \equiv \{g(z, y) \mid y \in \{1, 2, \cdots, K\}, z \in \mathbb{R}^m\}$. Given such a model we can in principle enumerate all unrestricted adversarial examples for a given classifier $f$, by finding all $z$ and $y$ such that $f(g(z, y)) \neq y$.

In practice, we can exploit different approximations of the ideal generative model to produce different kinds of unrestricted adversarial examples. Because of its reliable conditioning and high fidelity image generation, we choose AC-GAN [20] as our basic class-conditional generative model. In what follows, we explore two attacks derived from variants of AC-GAN (see pseudocode in Appendix B).

**Basic attack**    Let $g_\theta(z, y), c_\phi(x)$ be the generator and auxiliary classifier of AC-GAN, and let $f(x)$ denote the classifier that we wish to attack. We focus on *targeted* unrestricted adversarial attacks, where the attacker tries to generate an image $x$ so that $o(x) = y_\text{source}$ but $f(x) = y_\text{target}$. In order to produce unrestricted adversarial examples, we propose finding the appropriate $z$ by minimizing a loss function $\mathcal{L}$ that is carefully designed to produce high fidelity unrestricted adversarial examples.

We decompose the loss as $\mathcal{L} = \mathcal{L}_0 + \lambda_1 \mathcal{L}_1 + \lambda_2 \mathcal{L}_2$, where $\lambda_1, \lambda_2$ are positive hyperparameters for weighting different terms. The first component

$$\mathcal{L}_0 \triangleq -\log f(y_\text{target} \mid g_\theta(z, y_\text{source})) \tag{1}$$

encourages $f$ to predict $y_\text{target}$, where $f(y \mid x)$ denotes the confidence of predicting label $y$ for input $x$. The second component

$$\mathcal{L}_1 \triangleq \frac{1}{m} \sum_{i=1}^{m} \max\{|z_i - z_i^0| - \epsilon, 0\} \tag{2}$$

soft-constrains the search region of $z$ so that it is close to a randomly sampled noise vector $z^0$. Here $z = (z_1, z_2, \cdots, z_m), \{z_1^0, z_2^0, \cdots, z_m^0\} \overset{\text{i.i.d.}}{\sim} \mathcal{N}(0, 1)$, and $\epsilon$ is a small positive constant. For a good generative model, we expect that $x^0 = g_\theta(z^0, y_\text{source})$ is diverse for randomly sampled $z^0$ and $o(x^0) = y_\text{source}$ holds with high probability. Therefore, by reducing the distance between $z$ and $z^0$, $\mathcal{L}_1$ has the effect of generating more diverse adversarial examples from class $y_\text{source}$. Without this constraint, the optimization may always converge to the same example for each class. Finally

$$\mathcal{L}_2 \triangleq -\log c_\phi(y_\text{source} \mid g_\theta(z, y_\text{source})) \tag{3}$$

encourages the auxiliary classifier $c_\phi$ to give correct predictions, and $c_\phi(y \mid x)$ is the confidence of predicting $y$ for $x$. We hypothesize that $c_\phi$ is relatively uncorrelated with $f$, which can possibly promote the generated images to reside in class $y_\text{source}$.

Note that $\mathcal{L}$ can be easily modified to perform *untargeted* attacks, for example replacing $\mathcal{L}_0$ with $-\max_{y \neq y_\text{source}} \log f(y \mid g_\theta(z, y_\text{source}))$. Additionally, when performing our evaluations, we need to use humans to ensure that our generative model is actually generating images which are in one of the desired classes with high probability. In contrast, when simply perturbing an existing image, past work has been able to assume that the true label does not change if the perturbation is small. Thus, during evaluation, to test whether the images are legitimate and belong to class $y_\text{source}$, we use crowd-sourcing on Amazon Mechanical Turk (MTurk).

**Noise-augmented attack**    The representation power of the AC-GAN generator can be improved if we add small trainable noise to the generated image. Let $\epsilon_\text{attack}$ be the maximum magnitude of noise that we want to apply. The noise-augmented generator is defined as $g_\theta(z, \tau, y; \epsilon_\text{attack}) \triangleq g_\theta(z, y) + \epsilon_\text{attack} \tanh(\tau)$, where $\tau$ is an auxiliary trainable variable with the same shape as $g_\theta(z, y)$ and $\tanh$ is applied element-wise. As long as $\epsilon_\text{attack}$ is small, $g_\theta(z, \tau, y; \epsilon_\text{attack})$ should be indistinguishable from $g_\theta(z, y)$, and $o(g_\theta(z, \tau, y; \epsilon_\text{attack})) = o(g_\theta(z, y))$, i.e., adding small noise should preserve image quality and ground-truth labels. Similar to the basic attack, noise-augmented unrestricted adversarial examples can be obtained by solving $\min_{z, \tau} \mathcal{L}$, with $g_\theta(z, y_\text{source})$ in (1) and (3) replaced by $g_\theta(z, \tau, y_\text{source}; \epsilon_\text{attack})$.

One interesting observation is that traditional perturbation-based adversarial examples can also be obtained as a special case of our noise-augmented attack, by choosing a suitable $g_\theta(z, y)$ instead of the AC-GAN generator. Specifically, let $\mathcal{T}$ be the test dataset, and $\mathcal{T}_y = \{x \in \mathcal{T} \mid o(x) = y\}$. We can use a discrete latent code $z \in \{1, 2, \cdots, |\mathcal{T}_{y_\text{source}}|\}$ and specify $g_\theta(z, y)$ to be the $z$-th image in $\mathcal{T}_y$. Then, when $z^0$ is uniformly drawn from $\{1, 2, \cdots, |\mathcal{T}_{y_\text{source}}|\}$, $\lambda_1 \to \infty$ and $\lambda_2 = 0$, we will recover an objective similar to FGSM [2] or PGD [10].

## 4   Experiments

### 4.1   Experimental details

**Amazon Mechanical Turk settings**    In order to demonstrate the success of our unrestricted adversarial examples, we need to verify that their ground-truth labels disagree with the classifier's

Table 1: Attacking certified defenses on MNIST. The unrestricted adversarial examples here are untargeted and without noise-augmentation. Numbers represent success rates (%) of our attack, based on human evaluations on MTurk. While no perturbation-based attack with $\epsilon = 0.1$ can have a success rate larger than the *certified rate* (when evaluated on the training set) we are able to achieve that by considering a more general attack mechanism.

| Classifier \ Source | 0 | 1 | 2 | 3 | 4 | 5 | 6 | 7 | 8 | 9 | Overall | Certified Rate ($\epsilon = 0.1$) |
|---|---|---|---|---|---|---|---|---|---|---|---|---|
| Raghunathan et al. [16] | 90.8 | 48.3 | 86.7 | 93.7 | 94.7 | 85.7 | 93.4 | 80.8 | 96.8 | 95.0 | 86.6 | $\leq 35.0$ |
| Kolter & Wang [17] | 94.2 | 57.3 | 92.2 | 94.0 | 93.7 | 89.6 | 95.7 | 81.4 | 96.3 | 93.5 | 88.8 | $\leq 5.8$ |

predictions. To this end, we use Amazon Mechanical Turk (MTurk) to manually label each unrestricted adversarial example.

To improve signal-to-noise ratio, we assign the same image to 5 different workers and use the result of a majority vote as ground-truth. For each worker, the MTurk interface contains 10 images and for each image, we use a button group to show all possible labels. The worker is asked to select the correct label for each image by clicking on the corresponding button. In addition, each button group contains an "N/A" button that the workers are instructed to click on if they think the image is not legitimate or does not belong to any class. To confirm our MTurk setup results in accurate labels, we ran a test to label MNIST images. The results show that **99.6%** of majority votes obtained from workers match the ground-truth labels.

For some of our experiments, we want to investigate whether unrestricted adversarial examples are more similar to existing images in the dataset, compared to perturbation-based attacks. We use the classical A/B test for this, *i.e.*, each synthesized adversarial example is randomly paired with an existing image from the dataset, and the annotators are asked to identify the synthesized images. Screen shots of all our MTurk interfaces can be found in the Appendix E.

**Datasets**   The datasets used in our experiments are MNIST [25], SVHN [26], and CelebA [27]. Both MNIST and SVHN are images of digits. For CelebA, we group the face images according to female/male, and focus on gender classification. We test our attack on these datasets because the tasks (digit categorization and gender classification) are easier and less ambiguous for MTurk workers, compared to those having more complicated labels, such as Fashion-MNIST [37] or CIFAR-10 [38].

**Model Settings**   We train our AC-GAN [20] with gradient penalty [21] on all available data partitions of each dataset, including training, test, and extra images (SVHN only). This is based on the assumption that attackers can access a large number of images. We use ResNet [39] blocks in our generative models, mainly following the architecture design of [21]. For training classifiers, we only use the training partition of each dataset. We copy the network architecture from [10] for the MNIST task, and use a similar ResNet architecture to [13] for other datasets. For more details about architectures, hyperparameters and adversarial training methods, please refer to Appendix C.

## 4.2   Untargeted attacks against certified defenses

We first show that our new adversarial attack can bypass the recently proposed *certified defenses* [16, 17]. These defenses can provide a theoretically verified certificate that a training example cannot be classified incorrectly by any perterbation-based attack with a perturbation size less than a given $\epsilon$.

**Setup**   For each source class, we use our method to produce 1000 untargeted unrestricted adversarial examples without noise-augmentation. By design these are all incorrectly classified by the target classifer. Since they are synthesized from scratch, we report in Tab. 1 the fraction labeled by human annotators as belonging to the intended source class. We conduct these experiments on the MNIST dataset to ensure a fair comparison, since certified classifiers with pre-trained weights for this dataset can be obtained directly from the authors of [16, 17]. We produce untargeted unrestricted adversarial examples in this task, as the certificates are for untargeted attacks.

**Results**   Tab. 1 shows the results. We can see that the stronger of the two certified defenses, [17], provides a certificate for 94.2% of the samples in the MNIST test set with $\epsilon = 0.1$ (out of 1). Since our technique is not perturbation-based, 88.8% of our samples are able to fool this defense. Note this does not mean the original defense is broken, since we are considering a different threat model.

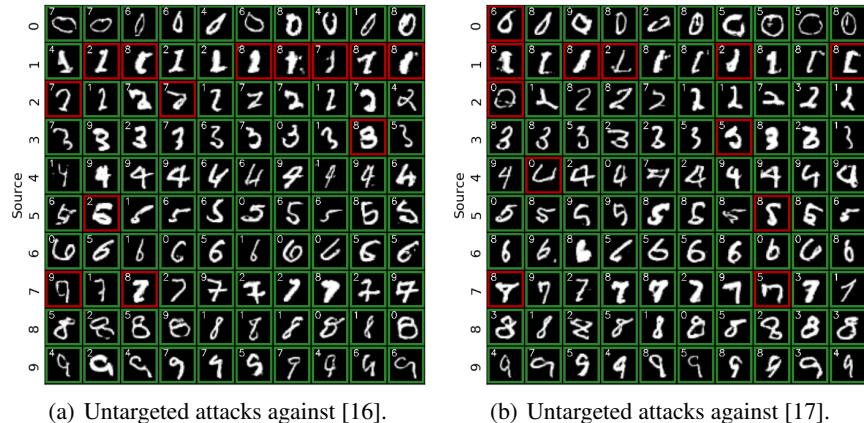

| (a) Untargeted attacks against [16]. | (b) Untargeted attacks against [17]. |

Figure 1: Random samples of untargeted unrestricted adversarial examples (w/o noise) against certified defenses on MNIST. Green and red borders indicate success/failure respectively, according to MTurk results. The annotation in upper-left corner of each image represents the classifier's prediction. For example, the entry in the top left corner (left panel) is classified as 0 by MTurk, and as a 7 by the classifier, and is therefore considered a success. The entry in the top left corner (right panel) is not classified as 0 by MTurk, and is therefore counted as a failure.

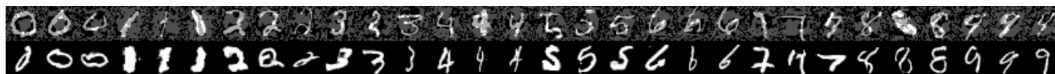

Figure 2: Comparing PGD attacks with high $\epsilon = 0.31$ (top) and ours (bottom) for [16] on MNIST. The two methods have comparable success rates of 86.0% (with $\epsilon = 0.31$) and 86.6% respectively. Our images, however, look significantly more realistic.

### 4.2.1 Evading human detection

One natural question is, *why not increase $\epsilon$ to achieve higher success rates?* With a large enough $\epsilon$, existing perturbation-based attacks will be able to fool certified defenses *using larger perturbations*. However, we can see the downside of this approach in Fig. 2: because perturbations are large, the resulting samples appear obviously altered.

**Setup**   To show this quantitatively, we increase the $\epsilon$ value for a traditional perturbation-based attack (a 100-step PGD) until the attack success rates on the certified defenses are similar to those for our technique. Specifically, we used an $\epsilon$ of 0.31 and 0.2 to attack [16] and [17] respectively, resulting in success rates of 86.0% and 83.5%. We then asked human annotators to distinguish between the adversarial examples and unmodified images from the dataset, in an A/B test setting. If the two are indistinguishable we expect a 50% success rate.

**Results**   We found that with perturbation-based examples [16], annotators can correctly identify adversarial images with a 92.9% success rate. In contrast, they can only correctly identify adversarial examples from our attack with a 76.8% success rate. Against [17], the success rates are 87.6% and 78.2% respectively. We expect this gap to increase even more as better generative models and defense mechanisms are developed.

### 4.3 Targeted attacks against adversarial training

The theoretical guarantees provided by certified defenses are satisfying, however, these defenses require computationally expensive optimization techniques and do not yet scale to larger datasets. Furthermore, more traditional defense techniques actually perform better in practice in certain cases. [33] has shown that the strongest non-certified defense currently available is the adversarial training technque presented in [10], so we also evaluated our attack against this defense technique. We perform this evaluation in the targeted setting in order to better understand how the success rate varies between various source-target pairs.

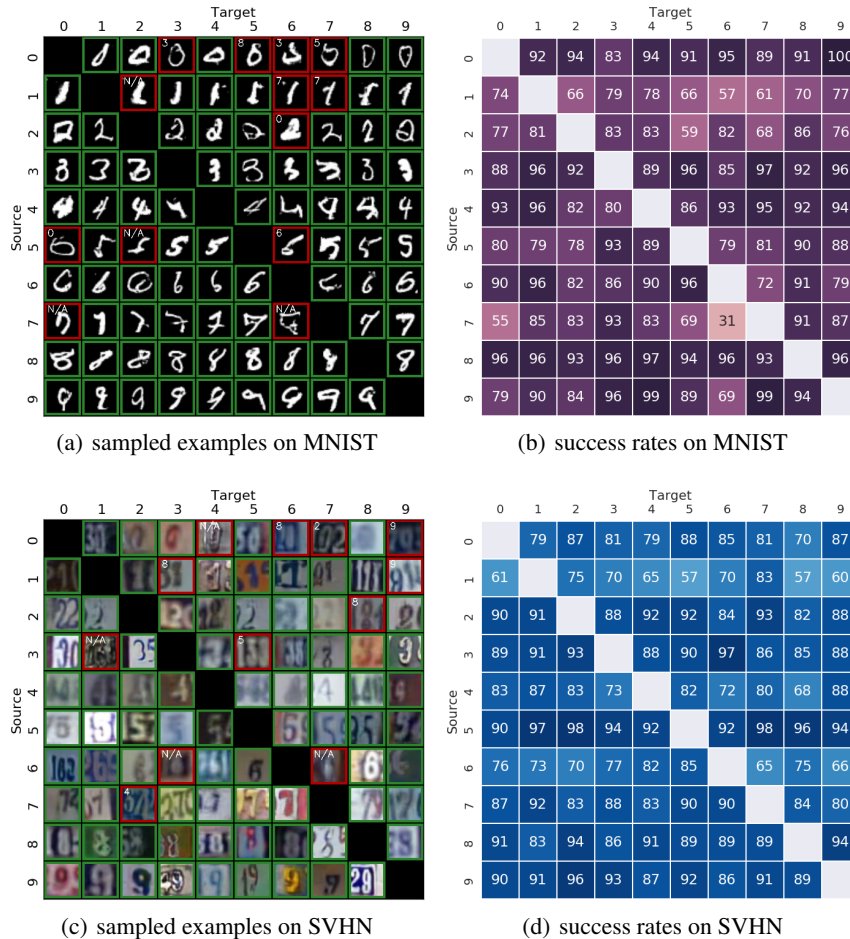

(a) sampled examples on MNIST

(b) success rates on MNIST

(c) sampled examples on SVHN

(d) success rates on SVHN

Figure 3: (a)(c) Random samples of targeted unrestricted adversarial examples (w/o noise). Row $i$ and column $j$ is an image that is supposed to be class $i$ and classifier predicts it to be class $j$. Green border indicates that the image is voted legitimate by MTurk workers, and red border means the label given by workers (as shown in the upper left corner) disagrees with the image's source class. (b)(d) The success rates (%) of our targeted unrestricted adversarial attack (w/o noise). Also see Tab. 2.

Table 2: Overall success rates of our targeted unrestricted adversarial attacks. Success rates of PGD are provided to show that the network is adversarially-trained to be robust. [†]Best public result.

| Robust Classifier | Accuracy (orig. images) | Success Rate of PDG | Our Success Rate (w/o Noise) | Our Success Rate (w/ Noise) | $\epsilon_{attack}$ |
|---|---|---|---|---|---|
| Madry network [10] on MNIST | 98.4 | 10.4[†] | 85.2 | 85.0 | 0.3 |
| ResNet (adv-trained) on SVHN | 96.3 | 59.9 | 84.2 | 91.6 | 0.03 |
| ResNet (adv-trained) on CelebA | 97.3 | 20.5 | 91.1 | 86.7 | 0.03 |

**Setup** We produce 100 unrestricted adversarial examples for each pair of source and target classes and ask human annotators to label them. We also compare to traditional PGD attacks as a reference. For the perturbation-based attacks against the ResNet networks, we use a 20-step PGD with values of $\epsilon$ given in the table. For the perturbation-based attack against the Madry network [10], we report the best published result [40]. It's important to note that the reference perturbation-based results are not directly comparable to ours because they are i) untargeted attacks and ii) limited to small perturbations. Nonetheless, they can provide a good sense of the robustness of adversarially-trained networks.

**Results** A summary of the results can be seen in Tab. 2. We can see that the defense from [10] is quite effective against the basic perturbation-based attack, limiting the success rate to 10.4% on

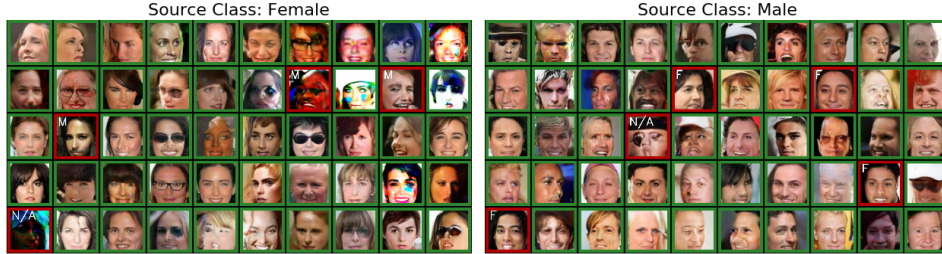

Figure 4: Sampled unrestricted adversarial examples (w/o noise) for fooling the classifier to misclassify a female as male (left) and the other way around (right). Green, red borders and annotations have the same meanings as in Fig. 3, except "F" is short for "Female" and "M" is short for "Male".

Table 3: Transferability of unrestricted adversarial examples on MNIST. We attack Madry Net [10] (adv) with our method and feed legitimate unrestricted adversarial examples, as verified by AMT workers, to other classifiers. Here "adv" is short for "adversarially-trained" (with PGD) and "no adv" means no adversarial training is used. Numbers represent accuracies of classifiers.

| Classifier<br>Attack Type | Madry Net [10]<br>(no adv) | Madry Net [10]<br>(adv) | ResNet<br>(no adv) | ResNet<br>(adv) | [16] | [17] |
|---|---|---|---|---|---|---|
| No attack | 99.5 | 98.4 | 99.3 | 99.4 | 95.8 | 98.2 |
| Our attack (w/o noise) | 95.1 | 0 | 92.7 | 93.7 | 77.1 | 84.3 |
| Our attack (w/ noise, $\epsilon_{\text{attack}} = 0.3$) | 78.3 | 0 | 73.8 | 84.9 | 78.1 | 63.0 |

MNIST, and 20.5% on CelebA. In contrast, our unrestricted adversarial examples (with or without noise-augmentation) can successfully fool this defense with more than an 84% success rate on all datasets. We find that adding noise-augmentation to our attack does not significantly change the results, boosting the SVNH success rate by 7.4% while reducing the CelebA success rate by 4.4%. In Fig. 3 and Fig. 4, we show samples and detailed success rates of unrestricted adversarial attacks without noise-augmentation. More samples and success rate details are provided in Appendix D.

## 4.4 Transferability

An important feature of traditional perturbation-based attacks is their transferability across different classifiers.

**Setup**   To test how our unrestricted adversarial examples transfer to other architectures, we use all of the unrestricted adversarial examples we created to target Madry Network [41] on MNIST for the results in Section 4.3, and filter out invalid ones using the majority vote of a set of human annotators. We then feed these unrestricted adversarial examples to other architectures. Besides the adversarially-trained Madry Network, the architectures we consider include a ResNet [39] similar to those used on SVHN and CelebA datasets in Section 4.3. We test both normally-trained and adversarially-trained ResNets. We also take the architecture of Madry Network in [10] and train it without adversarial training.

**Results**   We show in Tab. 3 that unrestricted adversarial examples exhibit moderate transferability to different classifiers, which means they can be threatening in a black-box scenario as well. For attacks without noise-augmentation, the most successful transfer happens against [16], where the success rate is 22.9%. For the noise-augmented attack, the most successful transfer is against [17], with a success rate of 37.0%. The results indicate that the transferability of unrestricted adversarial examples can be generally enhanced with noise-augmentation.

## 5   Analysis

In this section, we analyze why our method can attack a classifier using a generative model, under some idealized assumptions. For simplicity, we assume the target $f(x)$ is a binary classifier, where $x \in \mathbb{R}^n$ is the input vector. Previous explanations for perturbation-based attacks [2] assume that the score function $s(x) \in \mathbb{R}$ used by $f$ is almost linear. Suppose $s(x) \approx w^\intercal x + b$ and $w, x \in \mathbb{R}^n$ both have high dimensions (large $n$). We can choose the perturbation to be $\delta = \epsilon \, \text{sign}(w)$, so that

$s(x + \delta) - s(x) \approx \epsilon \cdot n$. Though $\epsilon$ is small, $n$ is typically a large number, therefore $\epsilon \cdot n$ can be large enough to change the prediction of $f(x)$. Similarly, we can explain the existence of unrestricted adversarial examples. Suppose $g(z, y) \in \mathbb{R}^n$ is an ideal generative model that can always produce legitimate images of class $y \in \{0, 1\}$ for any $z \in \mathbb{R}^m$, and assume for all $z^0$, $f(g(z^0, y)) = y$. The end-to-end score function $s(g(z, y))$ can be similarly approximated by $s(g(z, y)) \approx w_g^\mathsf{T} z + b_g$, and we can again take $\delta_z = \epsilon \operatorname{sign}(w_g)$, so that $s(g(z^0 + \delta_z, y)) - s(g(z^0, y)) \approx \epsilon \cdot m$. Because $m \gg 1$, $\epsilon \cdot m$ can be large enough to change the prediction of $f$, justifying why we can find many unrestricted adversarial examples by minimizing $\mathcal{L}$.

It becomes harder to analyze the case of an imperfect generative model. We provide a theoretical analysis in Appendix A under relatively strong assumptions to argue that most unrestricted adversarial examples produced by our method should be legitimate.

# 6   Related work

Some recent attacks also use more structured perturbations beyond simple norm bounds. For example, [42] shows that wearing eyeglass frames can cause face-recognition models to misclassify. [43] tests the robustness of classifiers to "nuisance variables", such as geometric distortions, occlusions, and illumination changes. [44] proposes converting the color space from RGB to HSV and shifting H, S components. [45] proposes mapping the input image to a latent space using GANs, and search for adversarial examples in the vicinity of the latent code. In contrast to our unrestricted adversarial examples where images are synthesized from scratch, these attacking methods craft malicious inputs based on a given test dataset using a limited set of image manipulations. Similar to what we have shown for traditional adversarial examples, we can view these attacking methods as special instances of our unrestricted adversarial attack framework by choosing a suitable generative model.

There is also a related class of maliciously crafted inputs named fooling images [46]. Different from adversarial examples, fooling images consist of noise or patterns that do not necessarily look realistic but are nonetheless predicted to be in one of the known classes with high confidence. As with our unrestricted adversarial examples, fooling images are not restricted to small norm-bounded perturbations. However, fooling images do not typically look legitimate to humans, whereas our focus is on generating adversarial examples which look realistic and meaningful.

Generative adversarial networks have also been used in some previous attack and defense mechanisms. Examples include AdvGAN [23], DeepDGA [47], ATN [48], GAT [49] and Defense-GAN [14]. The closest to our work are AdvGAN and DeepDGA. AdvGAN also proposes to use GANs for creating adversarial examples. However, their adversarial examples are still based on *small norm-bounded perturbations*. This enables them to assume adversarial examples have the same ground-truth labels as unperturbed images, while we use *human evaluation* to ensure the labels for our evaluation. DeepDGA uses GANs to generate adversarial domain names. However, domain names are arguably easier to generate than images since they need to satisfy fewer constraints.

# 7   Conclusion

In this paper, we explore a new threat model and propose a more general form of adversarial attacks. Instead of perturbing existing data points, our unrestricted adversarial examples are synthesized entirely from scratch, using conditional generative models. As shown in experiments, this new kind of adversarial examples undermines current defenses, which are designed for perturbation-based attacks. Moreover, unrestricted adversarial examples are able to transfer to other classifiers trained using the same dataset. After releasing the first draft of this paper, there has been a surge of interest in more general adversarial examples. For example, a contest [24] has recently been launched on unrestricted adversarial examples.

Both traditional perturbation-based attacks and the new method proposed in this paper exploit current classifiers' vulnerability to covariate shift [50]. The prevalent training framework in machine learning, Empirical Risk Minimization [51], does not guarantee performance when tested on a different data distribution. Therefore, it is important to develop new training methods that can generalize to different input distributions, or new methods that can reliably detect covariate shift [52]. Such new methods should be able to alleviate threats of both perturbation-based and unrestricted adversarial examples.

## Acknowledgements

The authors would like to thank Shengjia Zhao for reviewing an early draft of this paper. We also thank Ian Goodfellow, Ben Poole, Anish Athalye and Sumanth Dathathri for helpful online discussions. This research was supported by Intel Corporation, TRI, NSF (#1651565, #1522054, #1733686 ) and FLI (#2017-158687).

## Footnotes

[1]In previous drafts we called it Generative Adversarial Examples. We switched the name to emphasize the difference from [23]. Concurrently, the same name was also used in [24] to refer to adversarial examples beyond small perturbations.

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
