[Supplementary Material]

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

# A  Analysis of imperfect generators

In this section, we give one possible explanation for why in practice generative models can create adversarial examples that fool classifiers. We will now assume that generators not always generate legitimate images from the desired class. We will argue that, under some strong assumptions, when the classifier's prediction contradicts the generator's label conditioning, it is more likely for the classifier to make a mistake, rather than the generator generates an incorrect image.

In order to make our argument, we first need Proposition 1:

**Proposition 1.** *Let* $\mathbf{W} \in \mathbb{R}^{n \times m}$ *be a random matrix. Assume entries of* $\mathbf{W}$ *are mutually independent and bounded, i.e.,* $|w_{ij}| \leq K$ *for all* $i$ *and* $j$. *Then, with probability* $1 - \delta$, *the following bound holds*

$$\frac{1}{n} \max_{\|\Delta \mathbf{x}\|_\infty \leq \epsilon} \|\mathbf{W} \cdot \Delta \mathbf{x}\|_1 \leq 4\epsilon K \sqrt{\frac{m\left(m \log 2 + \log \frac{1}{\delta}\right)}{n}} + \epsilon K m. \tag{4}$$

Intuitively, Proposition 1 characterizes the robustness of a "typical" linear function as a function of its input and output dimensions. When the input dimension $m$ is fixed, the average maximum perturbation of output $\frac{1}{n} \max_{\|\Delta \mathbf{x}\|_\infty \leq \epsilon} \|\mathbf{W} \cdot \Delta \mathbf{x}\|$ is upper bounded by $4\epsilon K \sqrt{\frac{m\left(m \log 2 + \log \frac{1}{\delta}\right)}{n}} + \epsilon K m$, which decreases as $n$ gets greater. Similarly, when the output dimension $n$ is fixed, the average maximum perturbation is upper bounded by $O(m)$, which decreases as $m$ gets smaller. As long as the entries of the weight matrix are mutually independent and bounded, this relationship between robustness and dimensions persists.

Although not rigorously proven, we believe that a similar relationship holds for non-linear functions as well. Consider a non-linear function $\mathbf{f}(\mathbf{x})$, where $\mathbf{x} \in \mathbb{R}^m$ and $\mathbf{f}(\mathbf{x}) \in \mathbb{R}^n$. For each data sample $\mathbf{x}_i$, we can linearize $\mathbf{f}(\mathbf{x})$ at $\mathbf{x}_i$ to get $\mathbf{f}(\mathbf{x}) \approx \mathbf{J}_{\mathbf{x}_i}(\mathbf{x} - \mathbf{x}_i) = \mathbf{J}_{\mathbf{x}_i} \Delta \mathbf{x}$. We shall assume that the induced random matrix $\mathbf{J}_\mathbf{x} \in \mathbb{R}^{n \times m}, \mathbf{x} \sim P_x$ has bounded and mutually independent entries, and can therefore apply Proposition 1 to get the same robustness-dimension relationship.

In unrestricted adversarial attacks, we have a conditional generative model $\mathbf{g}(\mathbf{z}, y)$ that takes as input a random noise vector $\mathbf{z} \in \mathbb{R}^l$ and a label $y$ and generates an image from $\mathbb{R}^m$. The target classifier $\mathbf{f}(\mathbf{x})$ takes an image $\mathbf{x} \in \mathbb{R}^m$ and output scores from $\mathbb{R}^n$ that are subsequently used for classification. In practice, we usually have $l \ll m$ and $m \gg n$. From our discussion above, the generative model should be asymptotically more robust than the classifier. Therefore, when perturbing $\mathbf{z}$ such that $\mathbf{f}(\mathbf{g}(\mathbf{z}', y)), \|\mathbf{z}' - \mathbf{z}\|_\infty \leq \epsilon$ predicts an incorrect label (not $y$), it is more likely that $\mathbf{f}$ makes the mistake, rather than $\mathbf{g}$ not generating an image with label $y$. That could explain why we can construct legitimate unrestricted adversarial examples.

**Proof of Proposition 1**

*Proof.* Let $\mathbf{W} = (\mathbf{w}_1^\mathsf{T}, \mathbf{w}_2^\mathsf{T}, \cdots, \mathbf{w}_n^\mathsf{T})^\mathsf{T}$, where $\mathbf{w}_i$ represents the $i$-th row vector of $\mathbf{W}$. In order to bound $\max_{\|\Delta \mathbf{x}\|_\infty \leq \epsilon} \|\mathbf{W} \Delta \mathbf{x}\|_1$, we first bound $\|\mathbf{W} \Delta \mathbf{x}\|_1$ for any fixed vector $\Delta \mathbf{x}$ from the ball $\mathcal{B}_\epsilon := \{\mathbf{x} \mid \|\mathbf{x}\|_\infty \leq \epsilon\}$ and then apply union bound. Note that $\|\mathbf{W} \Delta \mathbf{x}\|_1$ can be written as the sum of $n$ terms, *i.e.*, $\|\mathbf{W} \Delta \mathbf{x}\|_1 = \sum_{i=1}^n |\mathbf{w}_i^\mathsf{T} \Delta \mathbf{x}|$, where each term $\mathbf{w}_i^\mathsf{T} \Delta \mathbf{x}$ can be bounded using McDiarmid's inequality [53]

$$\mathbb{P}\left(\sum_{j=1}^m w_{ij}(\Delta x)_j - M_i \geq \lambda\right) \leq \exp\left(\frac{-\lambda^2}{2mK^2\epsilon^2}\right), \tag{5}$$

where $M_i = \sum_{j=1}^m \mathbb{E}[w_{ij}(\Delta x)_j] \leq \epsilon K m$, according to the assumption.

From (5) we conclude the random variable $\mathbf{w}_i^\mathsf{T} \Delta \mathbf{x} - M_i$ is sub-Gaussian [54], hence

$$\mathbb{E}\left[\exp(s|\mathbf{w}_i^\mathsf{T} \Delta \mathbf{x} - M_i|)\right] \leq \exp\left(4s^2 mK^2\epsilon^2\right)$$

$$\Rightarrow \mathbb{E}\left[\exp\left(s \sum_{i=1}^n |\mathbf{w}_i^\mathsf{T} \Delta \mathbf{x} - M_i|\right)\right] \leq \exp(4s^2 nmK^2\epsilon^2).$$

With Markov inequality [53], we obtain

$$\mathbb{P}\left(\sum_{i=1}^{n}|\mathbf{w}_i^\intercal \Delta\mathbf{x} - M_i| \geq \lambda\right) \leq \exp(4s^2 nmK^2\epsilon^2 - s\lambda). \tag{6}$$

Because (6) holds for every $s \in \mathbb{R}$, we can optimize $s$ to get the tightest bound

$$\mathbb{P}\left(\sum_{i=1}^{n}|\mathbf{w}_i^\intercal \Delta\mathbf{x} - M_i| \geq \lambda\right) \leq \exp\left(-\frac{\lambda^2}{16nmK^2\epsilon^2}\right)$$

$$\Rightarrow \mathbb{P}\left(\|\mathbf{W}\Delta\mathbf{x}\|_1 \geq \lambda + \sum_{i=1}^{n}|M_i|\right) \leq \exp\left(-\frac{\lambda^2}{16nmK^2\epsilon^2}\right)$$

$$\Rightarrow \mathbb{P}\left(\|\mathbf{W}\Delta\mathbf{x}\|_1 \geq \lambda + K\epsilon nm\right) \leq \exp\left(-\frac{\lambda^2}{16nmK^2\epsilon^2}\right).$$

Now we are ready to apply union bound to control $\max_{\Delta\mathbf{x}\in\mathcal{B}_\epsilon}\|\mathbf{W}\Delta\mathbf{x}\|_1$. Although $\mathcal{B}_\epsilon$ is an infinite set, we only need to consider a finite set of vertices $\mathcal{V}_\epsilon := \{\mathbf{x} \mid x_i \in \{-\epsilon, \epsilon\}, \forall i = 1, 2, \cdots, m\}$. To see this, assume $\mathbf{x}^* = \arg\max_{\mathbf{x}\in\mathcal{S}_\epsilon}\|\mathbf{W}\mathbf{x}\|_1$ but $\mathbf{x}^* \notin \mathcal{V}_\epsilon$. Because $\mathcal{B}_\epsilon$ is a convex polytope with vertices $\mathcal{V}_\epsilon$, we have $\mathbf{x}^* = \sum_{i=1}^{2^m} \lambda_i \mathbf{v}_i$, where $\lambda_i \in [0,1]$, $\sum_{i=1}^{2^m} \lambda_i = 1$ and $\mathbf{v}_i$ denotes the $i$-th vertex in $\mathcal{V}_\epsilon$. By triangle inequality we have

$$\|\mathbf{W}\mathbf{x}^*\|_1 = \left\|\sum_{i=1}^{2^m} \lambda_i \mathbf{W}\mathbf{v}_i\right\|_1$$

$$\leq \sum_{i=1}^{2^m} \lambda_i \|\mathbf{W}\mathbf{v}_i\|_1$$

$$\leq \left(\sum_{i=1}^{2^m} \lambda_i\right) \max_{i\in[0,2^m]} \|\mathbf{W}\mathbf{v}_i\|_1$$

$$= \max_{i\in[0,2^m]} \|\mathbf{W}\mathbf{v}_i\|_1 \leq \|\mathbf{W}\mathbf{x}^*\|_1.$$

Let $\mathbf{v}^* := \arg\max_{\mathbf{v}_i, i\in[0,2^m]} \|\mathbf{W}\mathbf{v}_i\|_1$. From the above derivation we conclude $\|\mathbf{W}\mathbf{v}^*\|_1 = \|\mathbf{W}\mathbf{x}^*\|_1$ and therefore it is sufficient to only consider $\mathcal{V}_\epsilon$ for union bound:

$$\mathbb{P}\left(\max_{\Delta\mathbf{x}\in\mathcal{S}_\epsilon}\|\mathbf{W}\Delta\mathbf{x}\|_1 \geq \lambda + \epsilon Knm\right) = \mathbb{P}\left(\max_{\Delta\mathbf{x}\in\mathcal{V}_\epsilon}\|\mathbf{W}\Delta\mathbf{x}\|_1 \geq \lambda + \epsilon Knm\right)$$

$$\leq |\mathcal{V}_\epsilon|\mathbb{P}\left(\|\mathbf{W}\Delta\mathbf{x}\|_1 \geq \lambda + \epsilon Knm\right) \qquad \text{(Union Bound)}$$

$$\leq 2^m \exp\left(-\frac{\lambda^2}{16nmK^2\epsilon^2}\right).$$

In other words, with probability $1 - \delta$,

$$\max_{\Delta\mathbf{x}\in\mathcal{S}_\epsilon}\|\mathbf{W}\Delta\mathbf{x}\|_1 \leq 4K\epsilon\sqrt{mn\left(m\log 2 + \log\frac{1}{\delta}\right)} + \epsilon Knm,$$

and the statement of our theorem gets proved. $\qquad\qquad\square$

## B  Pseudocode

## C  Detailed experimental settings

**Datasets**  The datasets used in our experiments are MNIST [25], SVHN [26], and CelebA [27]. Both MNIST and SVHN are images of digits. MNIST contains 60000 28-by-28 gray-scale digits in the training set, and 10000 digits in the test set. In SVHN, there are 73257 32-by-32 images of house numbers (captured from Google Street View) for training, 26032 images for testing, and 531131 additional images as extra training data. For CelebA, there are 202599 celebrity faces, each of which has 40 binary attribute annotations. We group the face images according to female/male, and focus on gender classification. The first 150000 images are split for training and the rest are used for testing.

---

**Algorithm 1** Unrestricted Adversarial Example Targeted Attack

---
1: **procedure** ATTACK($y_\text{target}, y_\text{source}, f, \theta, \phi, \epsilon, \epsilon_\text{attack}, \lambda_1, \lambda_2, \alpha, T$)
2:　　Define $\mathcal{L}(z, \tau \, ; z^0, y_\text{target}, y_\text{source}, f, \theta, \phi, \epsilon, \epsilon_\text{attack}, \lambda_1, \lambda_2)$

$$\mathcal{L} = -\log f(y_\text{target} \mid g_\theta(z, y_\text{source}) + \epsilon_\text{attack}\tanh(\tau)) + \lambda_1 \cdot \frac{1}{m}\sum_{i=1}^{m}\max\{|z_i - z_i| - \epsilon, 0\}$$

$$- \lambda_2 \cdot \log c_\phi(y_\text{source} \mid g_\theta(z, y_\text{source}) + \epsilon_\text{attack}\tanh(\tau))$$

3:　　Initialize $x_\text{attack} \leftarrow \varnothing$
4:　　**while** $x_\text{attack} = \varnothing$ **do**
5:　　　Sample $\tau \sim \mathcal{N}(0, 1)$
6:　　　Sample $z^0 \sim \mathcal{N}(0, 1)$
7:　　　Initialize $z \leftarrow z^0$
8:　　　**for** $i = 1\ldots T$ **do**
9:　　　　Update $z \leftarrow z - \alpha \cdot \frac{\partial \mathcal{L}(z, \tau)}{\partial z}$　　　　　　　　　　　$\triangleright \alpha$ is the learning rate.
10:　　　　$\Delta \leftarrow \frac{\partial \mathcal{L}(z, \tau)}{\partial \tau}$
11:　　　　Update $\tau \leftarrow \tau - \alpha \cdot \frac{\Delta}{\|\Delta\|}$　　　$\triangleright$ We found gradient normalization to be effective
12:　　　**end for**
13:　　　$x \leftarrow g_\theta(z, y_\text{source}) + \epsilon_\text{attack}\tanh(\tau)$
14:　　　**if** $y_\text{target} = \arg\max_y p(f(x) = y)$ **then**
15:　　　　$x \leftarrow x_\text{attack}$
16:　　　**end if**
17:　　**end while**
18:　　**return** $x_\text{attack}$
19: **end procedure**

---

**Adversarial training**　　Regarding adversarial training, we directly use the weights provided by [10] for MNIST. For other tasks, we combine the techniques from [10] and [9]. More specifically, suppose pixel space is $[0, 255]$. We first sample $\epsilon$ from $\mathcal{N}(0, 8)$, take the absolute value and truncate it to $[0, 16]$, after which we use PGD with $\epsilon$ and iteration number $\lfloor \min(\epsilon + 4, 1.25\epsilon) \rfloor$ to generate adversarial examples for adversarial training. As suggested in [9], this has the benefit of making models robust to attacks with different $\epsilon$.

**Model architectures**　　For the AC-GAN architecture, we mostly follow the best designs tested in [21]. Specifically, we adapted their AC-GAN architecture on CIFAR-10 for our experiments of MNIST and SVHN, and used their AC-GAN architecture on 64×64 LSUN for our CelebA experiments. Since MNIST digits have lower resolution than CIFAR-10 images, we reduced one residual block in the generator so that the output shape is smaller, and reduced the channels of output from 3 to 1. The other components of AC-GAN, including architecture of the discriminator and auxiliary classifier, are all the same as described in [21].

For classifier architectures, we obtained networks and weights from authors of [10, 16, 17] so that we can be consistent with their papers. The ResNet architectures used for SVHN, CelebA and transferability experiments are shown in Tab. 5.

**Hyperparameters of attacks**　　Tab. 4 shows our hyperparameters of all unrestricted adversarial attacks used in this paper, following the same notations in arguments of Algorithm 1.

Table 4: Hyperparameters of unrestricted adversarial attacks. Notations are the same as in Algorithm 1. [*]Target class is "Male". [†]Target class is "Female".

| Datasets | Classifier | Targeted | Noise | $\lambda_1$ | $\lambda_2$ | $\epsilon$ | $\epsilon_{\text{attack}}$ | $\alpha$ | $T$ |
|---|---|---|---|---|---|---|---|---|---|
| MNIST | Madry Net [10] | Yes | No | 50 | 0 | 0.1 | 0 | 1 | 500 |
| MNIST | Madry Net [10] | Yes | Yes | 50 | 0 | 0.1 | 0.3 | 1 | 500 |
| MNIST | [16] | No | No | 100 | 0 | 0.1 | 0 | 10 | 100 |
| MNIST | [17] | No | No | 100 | 0 | 0.1 | 0 | 1 | 100 |
| SVHN | ResNet | Yes | No | 100 | 100 | 0.01 | 0 | 0.1 | 200 |
| SVHN | ResNet | Yes | Yes | 100 | 100 | 0.01 | 0.03 | 0.5 | 300 |
| CelebA[*] | ResNet | Yes | No | 100 | 100 | 0.001 | 0 | 1 | 200 |
| CelebA[*] | ResNet | Yes | Yes | 100 | 100 | 0.001 | 0.03 | 1 | 200 |
| CelebA[†] | ResNet | Yes | No | 100 | 100 | 0.1 | 0 | 0.1 | 200 |
| CelebA[†] | ResNet | Yes | Yes | 100 | 100 | 0.1 | 0.03 | 0.1 | 200 |

Table 5: ResNet Classifier Architecture. We set the base number of feature maps to $m = 4$ for MNIST and $m = 16$ for SVHN and CelebA.

| Name | Configuration | Replicate Block |
|---|---|---|
| Initial Layer | $3 \times 3$ conv. $m$ maps. $1 \times 1$ stride. | — |
| Residual Block 1 | batch normalization, leaky relu<br>$3 \times 3$ conv. $m$ maps. $1 \times 1$ stride<br>batch normalization, leaky relu<br>$3 \times 3$ conv. $m$ maps. $1 \times 1$ stride<br>residual addition | $\times 10$ |
| Resize Block 1 | batch normalization, leaky relu<br>$3 \times 3$ conv. $2m$ maps. $2 \times 2$ stride<br>batch normalization, leaky relu<br>$3 \times 3$ conv. $2m$ maps. $1 \times 1$ stride<br>average pooling, padding | — |
| Residual Block 2 | batch normalization, leaky relu<br>$3 \times 3$ conv. $2m$ maps. $1 \times 1$ stride<br>batch normalization, leaky relu<br>$3 \times 3$ conv. $2m$ maps. $1 \times 1$ stride<br>residual addition | $\times 9$ |
| Resize Block 2 | batch normalization, leaky relu<br>$3 \times 3$ conv. $4m$ maps. $2 \times 2$ stride<br>batch normalization, leaky relu<br>$3 \times 3$ conv. $4m$ maps. $1 \times 1$ stride<br>average pooling, padding | — |
| Residual Block 3 | batch normalization, leaky relu<br>$3 \times 3$ conv. $4m$ maps. $1 \times 1$ stride<br>batch normalization, leaky relu<br>$3 \times 3$ conv. $4m$ maps. $1 \times 1$ stride<br>residual addition | $\times 9$ |
| Pooling Layer | batch normalization, leaky relu, average pooling | — |
| Output Layer | 10 dense, softmax | — |

# D   Additional samples

(a) Extended plot for untargeted attacks against [16].

(b) Extended plot for untargeted attacks against [17].

Figure 5: (Extended plot of Fig. 1) Random samples of untargeted unrestricted adversarial examples (w/o noise) against certified defenses on MNIST. Green and red borders indicate success/failure respectively, according to MTurk results. The annotation in upper-left corner of each image represents the classifier's prediction.

Target

(a) success rates on MNIST

| Source \ Target | 0 | 1 | 2 | 3 | 4 | 5 | 6 | 7 | 8 | 9 |
|---|---|---|---|---|---|---|---|---|---|---|
| 0 |  | 89 | 93 | 93 | 92 | 86 | 85 | 97 | 92 | 96 |
| 1 | 70 |  | 72 | 82 | 82 | 73 | 72 | 82 | 85 | 80 |
| 2 | 81 | 79 |  | 92 | 89 | 78 | 78 | 80 | 98 | 86 |
| 3 | 85 | 90 | 93 |  | 82 | 85 | 74 | 87 | 80 | 93 |
| 4 | 84 | 94 | 77 | 85 |  | 91 | 93 | 92 | 84 | 81 |
| 5 | 74 | 78 | 70 | 89 | 88 |  | 79 | 85 | 89 | 89 |
| 6 | 82 | 95 | 85 | 89 | 91 | 96 |  | 79 | 90 | 79 |
| 7 | 59 | 78 | 77 | 83 | 84 | 75 | 47 |  | 86 | 78 |
| 8 | 95 | 95 | 92 | 97 | 91 | 95 | 95 | 92 |  | 97 |
| 9 | 84 | 80 | 86 | 89 | 94 | 82 | 72 | 97 | 93 |  |

(b) success rates on SVHN

| Source \ Target | 0 | 1 | 2 | 3 | 4 | 5 | 6 | 7 | 8 | 9 |
|---|---|---|---|---|---|---|---|---|---|---|
| 0 |  | 90 | 95 | 93 | 90 | 94 | 95 | 90 | 87 | 96 |
| 1 | 82 |  | 84 | 87 | 86 | 66 | 80 | 83 | 72 | 80 |
| 2 | 93 | 93 |  | 94 | 92 | 96 | 96 | 97 | 89 | 95 |
| 3 | 91 | 96 | 94 |  | 91 | 97 | 94 | 100 | 90 | 98 |
| 4 | 89 | 90 | 93 | 91 |  | 90 | 91 | 90 | 90 | 95 |
| 5 | 94 | 96 | 97 | 97 | 99 |  | 97 | 97 | 93 | 97 |
| 6 | 93 | 79 | 89 | 91 | 93 | 92 |  | 93 | 90 | 89 |
| 7 | 89 | 94 | 95 | 93 | 88 | 91 | 88 |  | 85 | 94 |
| 8 | 94 | 91 | 89 | 91 | 93 | 95 | 97 | 90 |  | 99 |
| 9 | 96 | 93 | 96 | 96 | 93 | 95 | 95 | 92 | 94 |  |

Figure 6: The success rates (%) of our targeted unrestricted adversarial attack with noise-augmentation.

Figure 7: (Extended plot of Fig. 3(a)) Random samples of targeted unrestricted adversarial examples (w/o noise) on MNIST. Green border indicates that the image is voted legitimate by MTurk workers, and red border means the label given by workers (as shown in the upper left corner) disagrees with the image's source class.

Figure 8: Random samples of targeted unrestricted adversarial examples (w/ noise) on MNIST. Green border indicates that the image is voted legitimate by MTurk workers, and red border means the label given by workers (as shown in the upper left corner) disagrees with the image's source class.

Figure 9: (Extended plot of Fig. 3(c)) Random samples of targeted unrestricted adversarial examples (w/o noise) on SVHN. Green border indicates that the image is voted legitimate by MTurk workers, and red border means the label given by workers (as shown in the upper left corner) disagrees with the image's source class.

Figure 10: Random samples of targeted unrestricted adversarial examples (w/ noise) on SVHN. Green border indicates that the image is voted legitimate by MTurk workers, and red border means the label given by workers (as shown in the upper left corner) disagrees with the image's source class.

Figure 11: (Extended plot of Fig. 4) Sampled unrestricted adversarial examples (w/o noise) for fooling the classifier to misclassify a female as male (left) and the other way around (right). Green, red borders and annotations have the same meanings as in Fig. 3, except "F" is short for "Female" and "M" is short for "Male".

Figure 12: Sampled unrestricted adversarial examples (w/ noise) for fooling the classifier to misclassify a female as male (left) and the other way around (right). Green, red borders and annotations have the same meanings as in Fig. 3, except "F" is short for "Female" and "M" is short for "Male".

# E MTurk web interfaces

The MTurk web interfaces used for labeling our unrestricted adversarial examples are depicted in Fig. 13. The A/B test interface used in Section 4.2.1 is shown in Fig. 14. In addition, Fig. 15 visualizes the uncertainty of MTurk annotators for labeling unrestricted adversarial examples, which indicates that more than 40%-50% unrestricted adversarial examples (depending on the dataset) get all of their 5 annotators agreed on one label.

(a) The MTurk web interface for MNIST and SVHN.

(b) The MTurk web interface for CelebA.

Figure 13: MTurk web interfaces for labeling unrestricted adversarial examples.

Figure 14: MTurk web interfaces for A/B test on MNIST.

Figure 15: The distribution of number of agreed votes for each image. For example, the red bar on top of 5 means around 55% unrestricted adversarial examples on CelebA dataset are voted the same label by all 5 MTurk annotators.