[Reviews · NeurIPS 2018]

Reviewer 1



****** EDIT: I am upgrading my review score in response to the authors' rebuttal. However, the similarities with Song et al should be made much more explicit, and the reviewers' other comments should also be taken into account. ******* This paper proposes a technique for generating adversarial examples that are not slight perturbations of an existing data sample. Rather, they are samples that are designed to look like a certain class to an observer, but get classified to a pre-specified, different class. The approach for doing so is to define a loss function that combines these two objectives (while also enforcing that the latent code vector should be near a randomly-selected initial vector). The authors show that this method is able to generate successful adversarial examples against schemes that are designed to protect against perturbation-based defenses. Overall, the paper seems to capture a more reasonable definition of adversarial examples; there is no inherent reason why adversarial examples should necessarily be based on an existing sample, and using GANs is an interesting way to decouple the adversarial example generation from the samples available in a dataset. Besides this insight, I didn’t feel like the ideas are particularly novel from a technical standpoint—the machine learning aspects of the paper are fairly standard. I think this work may be more interesting from a security standpoint than from a machine learning one. However, the authors did a fairly good job of evaluating their ideas (especially considering how subjective the problem is), and the paper was overall an enjoyable read. The paper states that using a larger epsilon is not effective because the samples are obviously tampered with. At least visually, I thought the samples in Figure 1 also looked tampered with (or at least clearly distorted). The experiment seems a little strange to me, because the samples with larger epsilon mainly appear to have a different background in Figure 2. This is a very easy thing to detect, but it doesn’t mean that the image looks substantially different from a hand-written digit. In fact, to me, the images just look like MNIST digits with a fuzzed background. Perhaps this experiment would have been more informative on a different dataset, like CIFAR? My concern is that because MNIST is so simple, it is easy to see these distortions in background, whereas they might not be as visible on a richer background. The paper is very well-written, though there are several typos that should be proofread.

Reviewer 2



After reviewing author response and other reviews, I'm slightly upgrading my score. I agree with the author response about the differences between the approach on Song et al, which MUST be a part of the revised paper. I acknowledge that adversarial images and adversarial domain names have different objectives. However, the authors should temper claims that this is the first paper to explore non L_p-norm constrained perturbations. The theme of the two papers are more alike than the authors rebuttal: generate a new object without constraints that can still evade a machine learning model. Creating a domain name that appears benign to human and domain classifier may nor may not be significantly easier than hallucinating an image that is misclassified by a convnet. This paper proposes generating adversarial examples (in a more general sense), from an AC-GAN. The key contribution appears to be the use of GANs to generate these general-case adversarial examples. It is clearly written. However, the originality is low. It collides with several previous works that the authors made no attempt to cite or references. Without searching, the reviewer is aware of these two recent examples (there are likely many others): (1) Song et al and colleagues authored similar work with the same motivation https://arxiv.org/abs/1801.02610 (2) Anderson et al demonstrated GANs for adversarially bypassing malicious domain detection, and also defined "adversarial examples" in this broader context https://arxiv.org/abs/1610.01969 Because of the lack of originality, this reviewer cannot recommend this paper for acceptance.

Reviewer 3



The paper proposes a new kind of adversarial attack to deep neural networks. Instead of perturbing an existing input image, the proposed method trains a generative adversarial network (GAN) and searches for generated image that will be misclassified. The paper is well written and idea is quite novel to me. I think it poses a new kind of threat for DNNs and opens a new research direction. L_p norm is not the best metric for computer vision, but so far it is the only focus in most existing work, due to its mathematical convenience. This work shows an attack beyond L_p constraint while still keeps the adversarial images legitimate. However, I have the following concerns and comments: 1. The result against certified defense and adversarial training is somewhat misleading, as they are proposed under a different threat model (i.e. perturbation-based attack). The authors should mention more explicitly that this is not a fair evaluation on these defense methods. 2. The authors use PGD in adversarial training, but I think a more natural way is to train on those generated adversarial examples. 3. The experiments are only on small and simple datasets. As it is hard to generate bigger and more complex images using GAN, it is unknown whether the proposed method can be used in other tasks (e.g., ImageNet classification).